# Who Cares about the Weather? Inferring Weather Conditions for Weather-Aware Object Detection in Thermal Images

Anders Skaarup Johansen [1,*,†], Kamal Nasrollahi [1,2], Sergio Escalera [1,3] and Thomas B. Moeslund [1]

1    Visual Analysis and Perception Lab, Department of Architecture, Design and Media Technology, Aalborg University, 9000 Aalborg, Denmark; kn@create.aau.dk (K.N.); sergio.escalera.guerrero@gmail.com (S.E.); tbm@create.aau.dk (T.B.M.)
2    Milestone Systems, 2605 Copenhagen, Denmark
3    Computer-Vision Center, Universitat de Barcelona, 08193 Bellaterra, Spain
*    Correspondence: asjo@create.aau.dk
†    Current address: Department of Architecture, Design and Media Technology, Aalborg University, Rendsburggade 14, 9000 Aalborg, Denmark.

**Featured Application: This work focuses on achieving a weather-agnostic approach for the real-world deployment of object recognition algorithms. Particularly object recognition conducted on thermal data exposed to long-term thermal drift.**

**Abstract:** Deployments of real-world object detection systems often experience a degradation in performance over time due to concept drift. Systems that leverage thermal cameras are especially susceptible because the respective thermal signatures of objects and their surroundings are highly sensitive to environmental changes. In this study, two types of weather-aware latent conditioning methods are investigated. The proposed method aims to guide two object detectors, (YOLOv5 and Deformable DETR) to become weather-aware. This is achieved by leveraging an auxiliary branch that predicts weather-related information while conditioning intermediate layers of the object detector. While the conditioning methods proposed do not directly improve the accuracy of baseline detectors, it can be observed that conditioned networks manage to extract a weather-related signal from the thermal images, thus resulting in a decreased miss rate at the cost of increased false positives. The extracted signal appears noisy and is thus challenging to regress accurately. This is most likely a result of the qualitative nature of the thermal sensor; thus, further work is needed to identify an ideal method for optimizing the conditioning branch, as well as to further improve the accuracy of the system.

**Keywords:** thermal; object detection; concept drift; conditioning; weather recognition

## 1. Introduction

Deploying thermal image recognition deep-learning models for the long-term analysis of a scene becomes increasingly difficult over time due to concept drift. Not only does the visual signature of the scene and objects within it change with seasons, but it also changes significantly between day and night. Thermal concept drift is an increasingly researched topic [1,2], and it is crucial for real-world deployments of thermal vision systems. Typically, the aim is focused on identifying distinct concept-drift factors or assuming the presence of distinct distributions. Establishing effective methods through which to combat concept drift is a vital component when deploying computer vision systems in real-world environments. Traditional evaluation methods do not provide accurate descriptions of the impact concept drift has on performance during long-term deployments [1]. Changes in contextual parameters, such as weather conditions, can be somewhat related to the degradation of performance observed with long-term concept drift [1]. For example, the relationship between the degradation of object detection models and changes in temperature and humidity is

statistically significant [1]. As weather conditions are somewhat correlated with changes in the visual appearance of captured thermal footage, they could be leveraged to guide the model toward learning two different representations: one that varies with the weather (weather-aware), and one that does not (weather-agnostic).

Multi-task learning has become an increasingly popular method for training generalized image-recognition models [2–7], but it mostly focuses on using auxiliary branches that are somewhat task-adjacent, i.e., where an intuitive connection to the primary task can be drawn. Each task contributes to converting the latent representation into a more generalized representation, which often increases performance for all tasks [4,6]. Given that the signals induced by the auxiliary tasks help to achieve a more robust representation, similar approaches could be leveraged to extract a contextually aware signal through auxiliary conditioning.

### 1.1. Estimating Weather

Directly leveraging weather information would require a vision system to directly infer weather conditions from the captured data [8,9]. By treating it as a classification problem, deep learning methods have shown great promise at classifying categories of weather [8–10]. This shows that a weather signal can be somewhat extracted from single images and categorized into distinct classes. Most weather classification approaches focus on the binary classification of distinct weather conditions (i.e., cloudy, sunny, raining, etc.), and they lack the granularity observed during long-term deployment. To address this, datasets like Rain Fog Snow (RFS) [11] and Multi-class Weather Dataset (MWD) [12] propose treating it as a multi-label classification problem in order to capture the ambiguity between different weather phenomena and transitive weather conditions [11,12]. When estimating weather conditions from a single image, all regions are not created equal [12,13]: thus, some methods isolate predetermined regions, such as the sky [9,14], or leverage region-proposal networks [12,13] to extract region-specific features.

### 1.2. Adapting to Weather

Adverse weather conditions, particularly those that are not present in the training dataset, present a real challenge for deployed computer vision systems that are exposed to the weather. Typically, approaches to concept drift rely on detecting drift, and then adapting accordingly [15,16]. With deep-learning-based approaches, this typically means a general system will be trained to establish a baseline, and this would then subsequently be exposed to unseen data. Depending on the task, an evaluation metric will be used as a method to detect drift [17]. When adapting to weather-related drift, some have tried to remove the distracting elements directly [18–22]; train weather-agnostic models by simulating various weather conditions and including that in the training loop [23–26]; or train several models in an ensemble and leveraging a weighted approach to determine the final prediction [27–30]. Moreover, in situations where an unsatisfactory amount of variation can be captured in the training data, continual-learning [16,31] or domain-adaptation [15,16] approaches are often leveraged in an attempt to obtain a consistent performance as the visual appearance of the context changes.

### 1.3. Leveraging Metadata for Recognition

In recent years, optimization, including auxiliary optimization, tasks has been shown to greatly improve the performance of the downstream task, whether used as a pre-text task (as often seen with vision transformers [32–35]) or as jointly optimized with the downstream task [33,36]. Using auxiliary tasks to guide a primary task by introducing aspects that cannot be properly captured in the downstream task's optimization objective has shown great promise in improving the performance and generalization of a downstream task [2,11]. Depending on the model architecture and desired purpose of this weather-conditioned representation, it can be leveraged as a constraining parameter so as to enforce the inclusion of the auxiliary representation directly, as shown in task-conditioned studies You Only Look

Once (YOLO) [2], thereby forcing the network to adjust to being aware of the contextual information induced. Alternatively, the auxiliary representation could be seen as purely supplemental information, which potentially consists of redundant elements, and—as such—should only be leveraged to indirectly guide the network.

### 1.4. Qualitative vs. Quantitative Thermal Cameras

Thermal cameras work by capturing the amount of infrared radiation emitted by objects within a scene. Though they all aim to capture the same type of information (namely heat), there are two types of thermal cameras. First is qualitative thermography (sometimes referred to as "relative thermal imaging"), where the goal is to show the relative differences of infrared radiation throughout the camera's field of view. These are often used for inspection and security purposes as they often provide distinct contrast between colder and hotter elements in the field of view, regardless of absolute temperature. Second is quantitative thermography (sometimes referred to as "absolute thermal imaging"), where each sampling point in the field of view is mapped to an absolute temperature measurement. This enables accurate capture of absolute thermal differences between elements in the scene, as well as a consistent visual response for any thermal signature. In recent years, advances in thermal imaging technology have made the use of thermal cameras increasingly popular, either in isolation or in conjunction with traditional Closed Circuit TeleVision (CCTV)-cameras. Quantitative thermal cameras could be seen as the ideal solution as they provide essentially the same functionality as qualitative thermal cameras but with the added benefit of accurate thermal readings. The technology required to construct an absolute thermograph is significantly more complicated than that of relative thermography, and—as such—is much more costly to produce and purchase. For the purpose of many tasks, the absolute temperature readings are redundant for the purpose of the thermal camera, and as such do not justify the cost, thus making qualitative thermal cameras much more common in deployed vision systems.

In this paper, a methodology is detailed that employs predicting continuous weather-related meta-variables for auxiliary guidance. In addition, an overview of the largest thermal object detection dataset in a stationary surveillance context is provided. The Long-term Thermal Drift (LTD) [1] dataset will be used to evaluate and train models as it contains both object-centric annotations, as well as fine-grained weather information for each sample. Further methods describe how fine-grained weather prediction can be leveraged to condition the network during training to guide the network to become weather-aware. Specifically, this will be divided into direct and indirect conditioning methods. Finally, this is followed by a discussion of extensive experiments, which evaluated the impact of the aforementioned methodology (conditioned on temperature, humidity, and time-of-day) and the impact on performance metrics with respect to the respective weather conditions. While the analysis does not show a direct improvement in accuracy metrics, it does show that auxiliary conditioning in this way allows the networks to extract and somewhat model the underlying weather signal.

## 2. Methodology

As weather conditions greatly alter the visual appearance of thermal imagery, infusing this knowledge into the models during training could serve as useful guidance toward weather-aware object detectors that outperform naively trained models. To achieve this, a model must be able to accurately estimate the current weather conditions. While more granular prediction schemes have become available for weather estimation, the methods for predicting weather conditions found in the literature are still predominantly performed using a binary scheme. Using a binary scheme as a conditioning method assumes that there is a fixed amount of distribution to the model. In an uncontrolled environment, this is a potentially false assumption as unknown variables could introduce noise to the signal that would make it difficult to distinguish the ground truth close to the bin edges [37,38]. This is potentially further exacerbated when processing thermal video from cameras with a relative

internal thermograph. As detailed in Section 1.4, the prevalence of relative thermal cameras make them a promising modality to investigate, particularly for a real-world context.

Prior work in auxiliary guidance have primarily leveraged multiple modalities [39–43], or have focused on guidance in the RGB domain [4,6,44–46]. To our knowledge, the only existing work that performs auxiliary conditioning for thermal-only object detection is the task-conditioned study of YOLO [2]. The authors leveraged a direct conditioning approach (detailed in Section 2.3) on the KAIST Multispectral Pedestrian Detection (KAIST) dataset [47], and managed to detect the objects overlooked by the baseline. However, the KAIST dataset contains thermal images from an absolute thermal camera, thereby resulting in a fairly similar thermal signature to that of pedestrians (as seen in Figure 1). Additionally, it can be observed in Figure 1 that this consistent thermal signature results in very low contrast between objects in certain scenes. Contrast-enhanced versions are available in Figure A1.

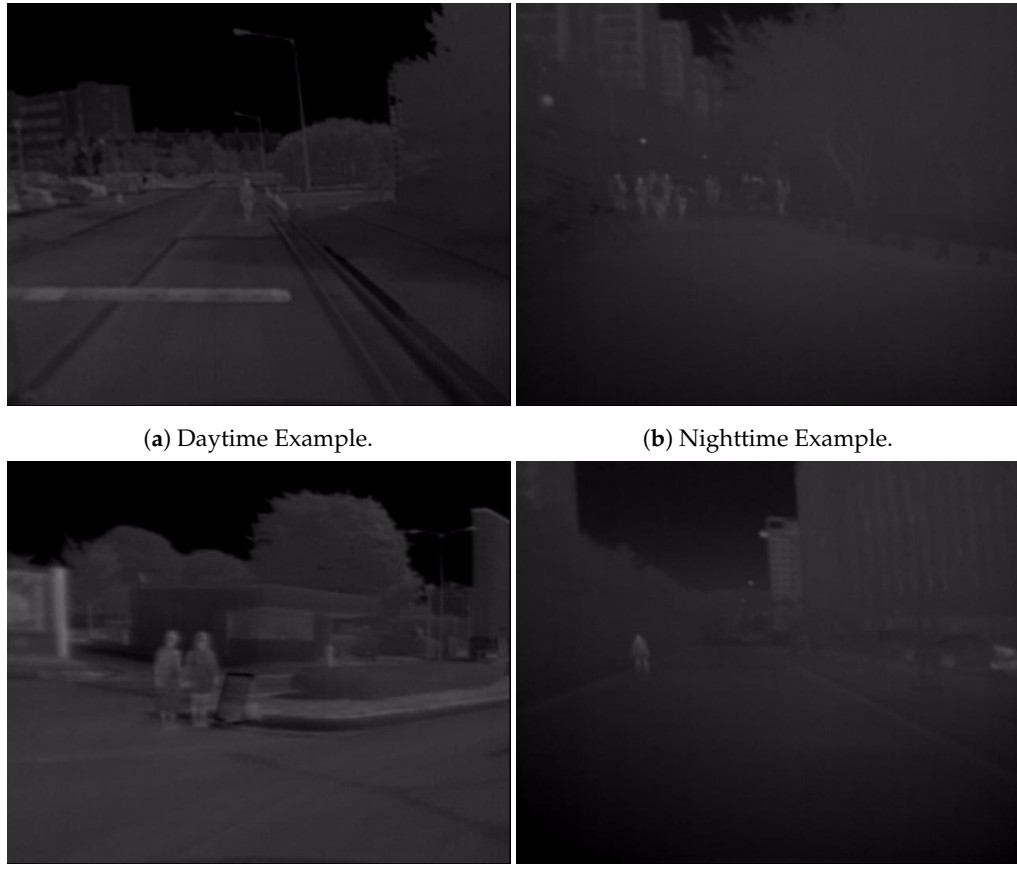

(**a**) Daytime Example.   (**b**) Nighttime Example.

(**c**) Daytime Example.   (**d**) Nighttime Example.

**Figure 1.** Examples of thermal images in the KAIST [47] dataset, where a similar thermal signature of people can be observed at different times of the day. This is due to the use of quantitative thermography, as well as due to the limited periods of captured data.

In this section, an overview of the object-centric annotations of the LTD-Dataset [1] and the associated meta-data are detailed. Furthermore, the method proposed in [2] can be adapted for a prediction of a continuous auxiliary variable. Finally, the architecture of a direct conditioning approach (similar to [2]), as well as an indirect conditioning approach that uses a State of the Art (SotA) transformer-based model, is detailed.

## 2.1. Dataset

In the original LTD dataset benchmark [1], and the subsequent ChaLearn: seasons in drift challenge [48], the performance impact concept drift has on object detectors is correlated with the absolute change in detection accuracy across different concept drift-related meta-variables (most notably temperature, humidity, and time of day). Subsequently, the

dataset has been extended with additional object-centric annotations. The dataset was uniformly sampled with a 0.5 frames per second sample rate (i.e., one frame every other second), resulting in over 900,000 images with over 6,000,000 annotated objects.

As can be seen in Figure 2, the thermal signature of people varies significantly more in the LTD dataset, as does the contrast between the objects and background.

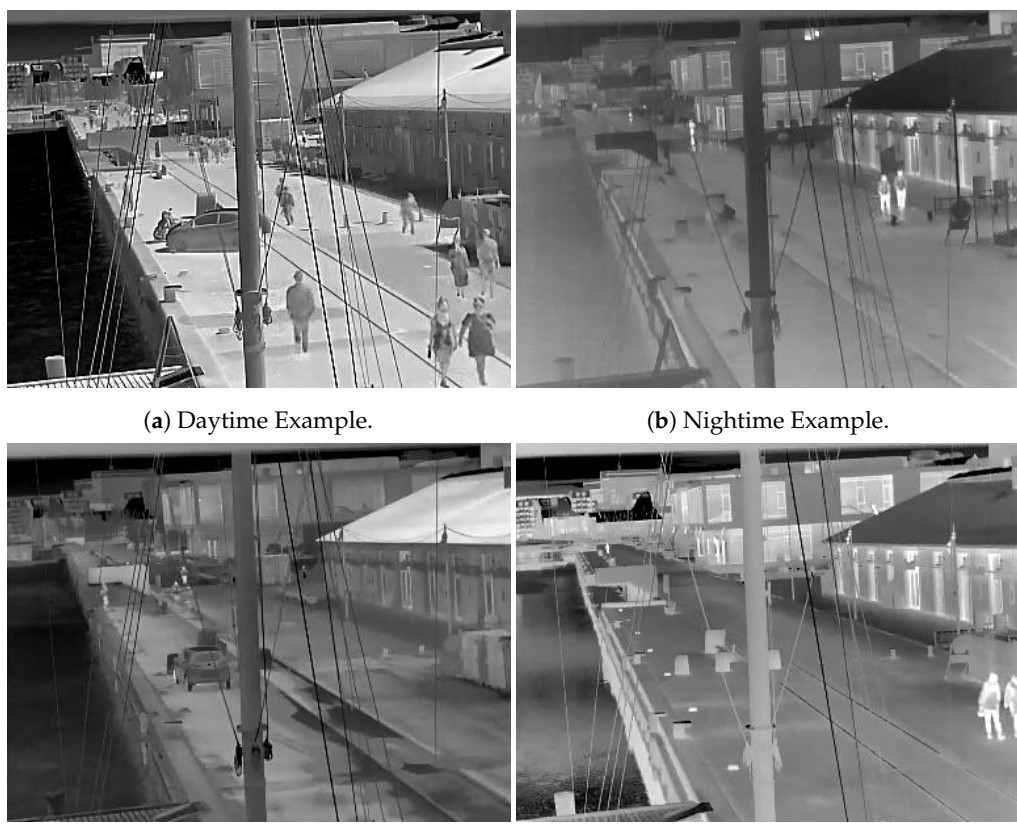

(**a**) Daytime Example.　　　　　　　　　　(**b**) Nightime Example.

(**c**) Daytime Example　　　　　　　　　　(**d**) Nighttime Example

**Figure 2.** Examples of images containing people from the LTD Dataset [1], where drastically different thermal signatures for objects can be observed. This is due to the use of qualitative thermography, as well as due to the dataset spanning 9 months.

The LTD dataset is captured in a real-world, unconstrained context, and is thus susceptible to associated biases, such as skewed object distribution (seen in Figure 3a); frames without objects of interest; highly varied object densities; and the uneven distribution of weather conditions (as seen in Figure 4a–c). The objects in the dataset are categorized into four classes—namely people, bicycles, motorcycles, and vehicles—spanning various shape and size configurations. Furthermore, the sizes of objects are affected by the camera being suspended roughly 6 m above the ground and being aimed downwards, which results in most objects appearing smaller than they, in reality, are (as seen in Figures 3b and A2). However, this phenomenon should be expected for deployment in a real-world security context.

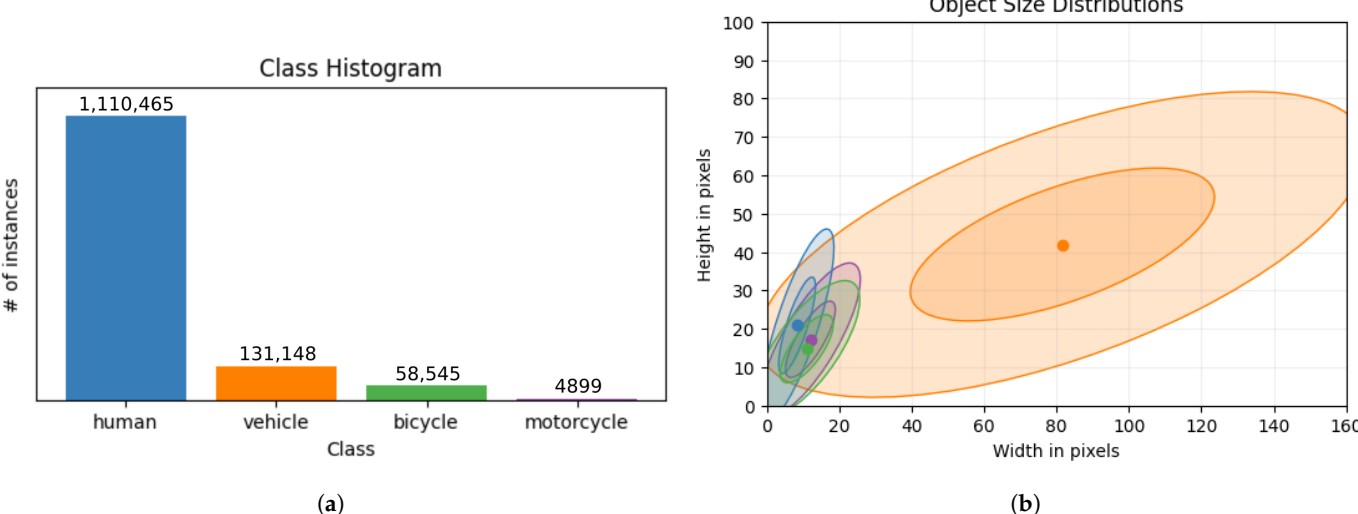

(**a**)    (**b**)

**Figure 3.** In (**a**), the total amount of instances from each of the given classes can be observed. Meanwhile, in (**b**), the average size for each class visualized in (**a**) can be seen marked as a dot. The surrounding circles visualize the span of object's height and width, as well as the 1 and 2 standard deviations, respectively.

Furthermore, as shown in Figure 3b, while each class has its own unique distribution, the distributions predominantly contain very small objects, with the exception of the vehicle class. This adds an additional degree of difficulty as most object detectors tend to struggle with smaller objects [32,49,50]. Additionally, a visualization of class-specific object size distribution can be found in Figures A4 and A5, thereby showing object sizes across the entire dataset, as well as in the training and validation splits.

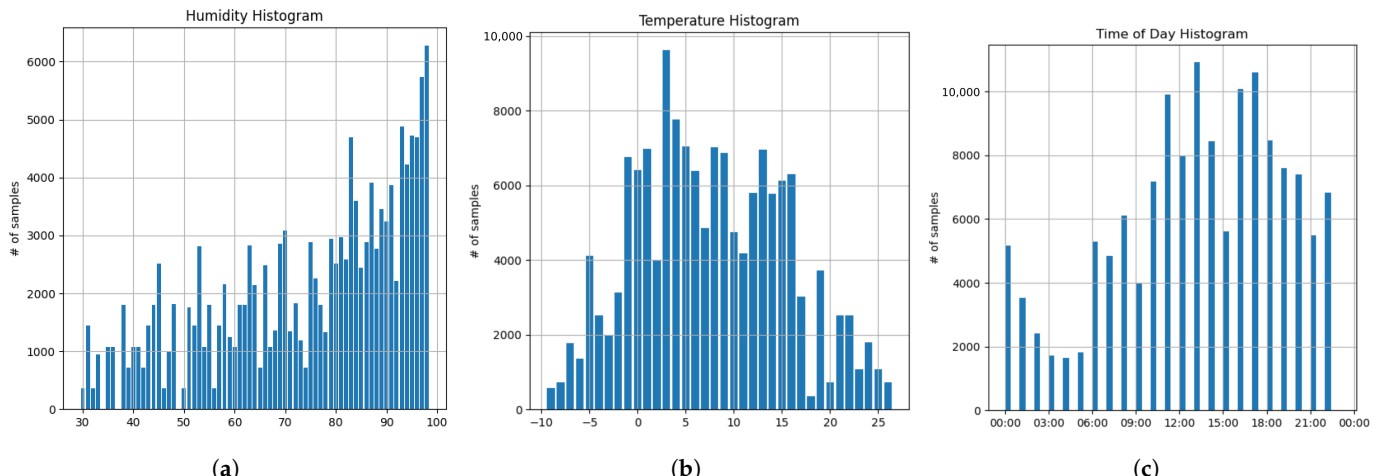

(**a**)    (**b**)    (**c**)

**Figure 4.** Histograms showing the distribution of meta-variables across the entire dataset. Visualizing the number of samples of a given humidity (**a**), temperature (**b**), and time-of-day (**c**).

Performance on the LTD [1] and KAIST [47] datasets are typically measured as the mean of the Average Precision (AP) across each class, denoted as mean Average Precision (mAP). Furthermore, a detection can only be considered a true positive, with respect to the joint area of both prediction and ground truth, if the predicted area intersecting with the ground truth is greater than a selected threshold. This is referred to as an Intersection over Union (IoU) threshold. Most commonly, the Pascal Visual Object Challenge (VOC) [51]- and Microsoft Common Objects in COntext (MS COCO) [52] variants of mAP are used.

*2.2. From Discrete to Continuous Meta-Prediction*

In the KAIST dataset, the data falls into two distinct categories, daytime and nighttime. However, in a real-world deployment, the system would observe a gradual change between daytime and nighttime, which is not accurately represented by such a binary grouping. However, in the LTD dataset [1], each clip has an extensive, highly granular set of metadata. This allows us to evaluate the impact of auxiliary task conditioning in real-world scenarios with more diverse samples. Furthermore, the distillation of this knowledge through conditioning could serve to make the model weather-aware, thus allowing it to adjust to weather-induced visual changes.

In [2], the authors proposed guiding the conditioning branch with a binary classification head (i.e., classifying day or night). However, to perform fine-grained, continuous weather prediction, the auxiliary optimization task and loss must be adjusted accordingly. The problem with binary classification is that it treats all false positives equally, regardless of the magnitude of the error. For continuous classifications, however, the severity of the misclassification can be assessed by determining the absolute difference between the prediction and the ground truth. Naively, an $L1$ loss can be used to punish/reward the network based on the difference in absolute distance. However, due to the data being captured by a relative thermal camera, identical visual appearance cannot be guaranteed between calibrations. During the capture of the data for the LTD dataset, the camera would routinely undergo automatic calibration, thus resulting in an inconsistent profile over time. This induces a noise signal that could result in the optimization converging toward a global mean rather than an acceptable guess. We combat this by employing an exponential $L1$ that is tuned to allow a pre-determined degree of deviation, and this is conducted before approaching the values of the primary task loss or losses.

We employed a variation of an exponential $L1$ loss, inspired by a naive exponential $L1$ loss [53], with a novel exponent-based soft reduction for losses lower than a specified deviation. Denoted as $k$, the accepted variance reduces the punishment as long as the absolute $L1$ distance to the ground truth is closer than $k$.

$$L1_e(x, y) \equiv L \equiv \{l_1, \ldots, l_N\}, l_n \equiv |x_n - y_n|^{\frac{k}{|x_n - y_n|}} \tag{1}$$

$L1_e(x, y)$ denotes the total loss $L$ given the prediction $x$ and ground truth $y$. The sum of losses in a given batch of size $N$ is the sum of the individual loss' $l_n$ at each position $n$. The individual losses $l_n$ are normalized with respect to $k$ over the absolute $L1$ distance, thus resulting in an exponent <1 for samples within the accepted deviation and an exponent >1 for samples exceeding the accepted deviation.

When establishing the baseline performance for each model, the minimum, maximum, and standard deviation of the primary loss was noted down for the final epoch. A corresponding $k$ that would approach the expected loss values of the primary task was selected. This is where the resulting weighting of the auxiliary in the optimization process would be approximately equal to that of the primary task at the border of the desired deviation $k$, all the while exponentially increasing when deviating further from the allowed $k$.

Due to the thermal images of the LTD dataset being recorded with a relative thermal camera, the visual appearance of a scene might be slightly different, even under similar meta-conditions. Thus, predicting the exact values from visual data would be an ill-posed problem, as any given state inherits some degree of variance from the calibration of the relative thermograph.

To evaluate the accuracy of the auxiliary module, we reported the Mean Average Error (MAE), Mean Average Percentage Error (MAPE), and Standard Deviation (Std.) to provide insights into the average error across all weather-related predictions; the percentage average error to access the magnitude of the error; and the standard deviation of the predictions around the target ground truth.

### 2.3. Direct Conditioning

In [2], the authors proposed a method of directly conditioning the latent representation of each predictive branch through a conditioning layer. The conditioning element is part of an auxiliary classification network, one that is aimed at predicting whether the given sample belongs to the daytime distribution or the nighttime distribution. The latent representation used for this auxiliary prediction is derived from an intermediate representation of the entire image. Thus, the representation must be able to extract a notion of day and night, which can make the network 'aware' and adapt accordingly.

In task-conditioned YOLO [2], the overall mAP does not improve significantly at higher IoUs; however, the weather-conditioned network shows a reduction in object Miss-Rate (MR). By directly conditioning the intermediate representation, the network is forced to directly incorporate the weather information in its semantically rich representation. We employ the original implementation of the YOLOv5 model. As shown in Figure 5a, the standard YOLO architecture is extended with an auxiliary branch, and it is extracted from one of the early stages of the feature extractor. Subsequently, a series of fully connected layers condense the representation to feed it to a prediction head. In addition, the prediction head produces a single value that is regressed following the exponential $L1$ loss described in Section 2.2. Individual fully connected layers feed the representation to the conditioning layer in the different stages of the network, and this occurs prior to the given stage's prediction head. The conditioning layer (shown in Section 2.3) takes in the set of feature maps, as well as an element-wise multiplication and summation with separated auxiliary representations $\alpha_n$ and $\beta_n$, respectively. Thus, the latent representation used for predicting the weather-related variable is directly enforced into the intermediate representation used by the object detector, which forces it to adjust accordingly.

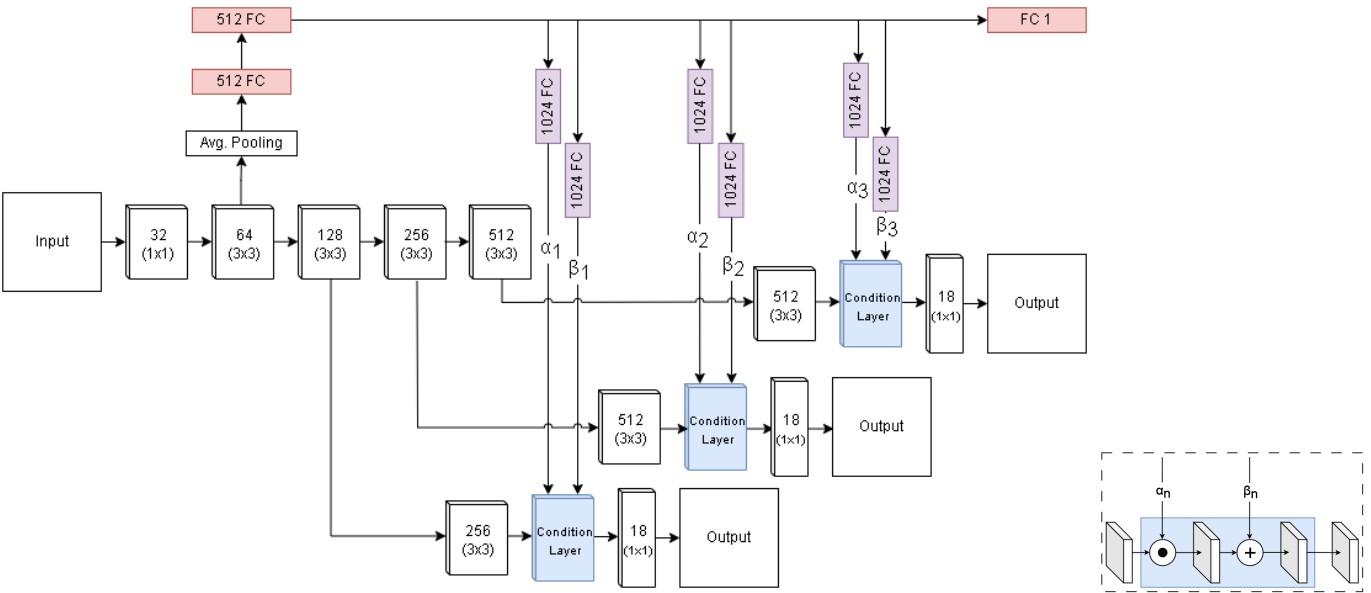

(**a**) YOLOv5 with a conditioning branch.　　　　　　　　　　　　　　(**b**) Conditioning layer

**Figure 5.** YOLO-styled conditioning network (Figure 5) and the internals of the conditioning layer (Section 2.3). Red, purple, and blue denote the auxiliary branch, mapping, and conditioning layers, respectively. The term '1024 FC' denotes the fully connected layers that project the latent representation to a single $1 \times 1024$ tensor. These latent representations are denoted as $\alpha_n$ and $\beta_n$ when inputted to the conditioning layer portrayed in Section 2.3. The white blocks denote convolutional bottleneck layers, where the channel depth (i.e., 32, 64, etc.) is denoted above the kernel size of the convolution.

### 2.4. Indirectly Imposed Conditioning

Vision transformers have proven to effectively leverage global reasoning to solve various vision tasks. By calculating an all-to-all affinity mapping (self-attention) between input

elements (tokens), transformers can effectively relate to elements, even when they belong to separate modalities. For classification, this is often employed with an additional learnable element, which is then mapped to a prediction head. The repeated self-attention allows the classification token to extract information from the entire input without directly imposing changes to other inter-token relationships. DEtection TRansformer (DETR) [32,49,54] is a common state-of-the-art transformer-based object detector. Subsequent variants have been shown to greatly improve convergence and the stability of optimization [49,54], and this is achieved by extending the deformable-DETR [54] with a learnable classification token and using the encoding of the classification token to perform predictions of the auxiliary task, namely weather condition prediction. While the optimization could potentially drive the transformer to learn embeddings that are optimized toward affinity with the classification token, the network should be able to disregard weather-related embeddings in cases where the weather does not provide any significant optimization benefit. Unlike the directly-imposed approach, the network could learn to dynamically disregard regions of the image that do not provide contextual information.

Inspired by the use of a classification token, i.e., '[CLS]', which was proposed in the original BERT [55] paper, we included an additional token with every input sample, and this was propagated through every encoder layer of the transformer. In this way, the global information from a given sample can be continuously aggregated in a single representation. Prior to reaching the decoder layers, the [CLS] token is separated and passed to an auxiliary branch (as seen in Figure 6). The auxiliary branch consists of a series of fully connected layers (sizes; 512, 512, and 1), thus acting as a mapping from latent representation to a single value that can be regressed. This method allows for the integration of visual information in the weather-related [CLS] token, which is achieved by extracting information directly from the visual input without directly enforcing the subsequent decoder to process the weather-related information. This disjointed approach theoretically should allow the network to disregard information that does not contribute to the weather prediction, unlike the direct conditioning approach.

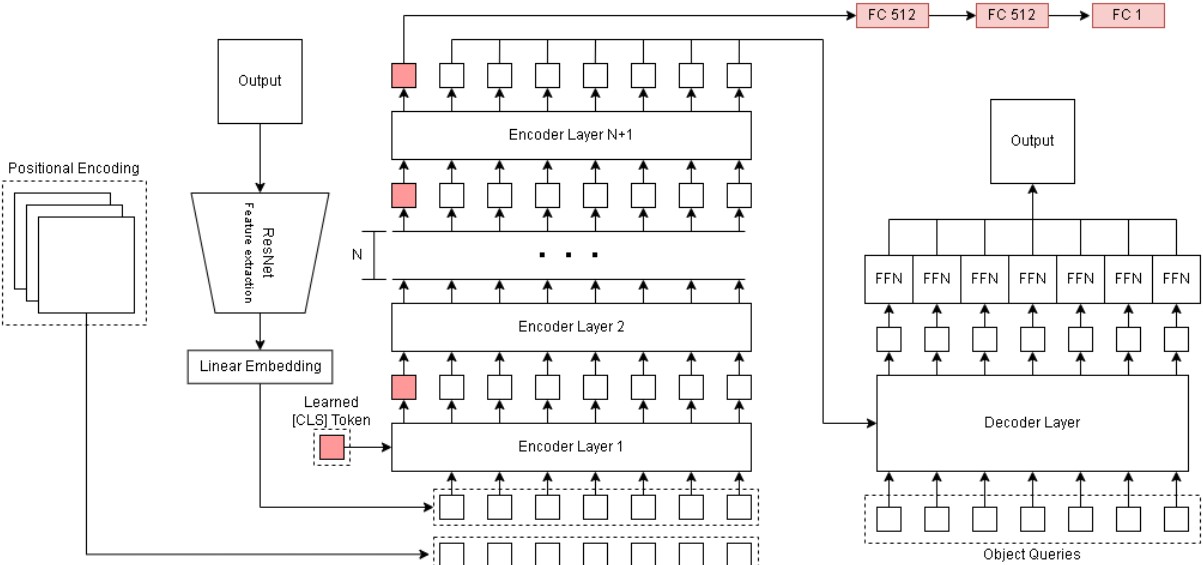

**Figure 6.** DETR-style transformer network with indirect conditioning. Red modules and tokens denote components used for auxiliary prediction. 'FC 512' denotes the fully connected layers that project the latent representation to a single $1 \times 512$ tensor.

## 3. Results

### 3.1. Experimental Setting

To establish a baseline, the models were trained as described in their respective papers and implementations. Since none of the models contained a thermal variant, they were all

trained from scratch. This required increased training time in order to expect convergence as "standard" configurations are implemented by loading an image-net pre-trained feature extraction network. For this reason, we set the maximum allowed epochs to 250 for all models. The batch size for all models was also set to 8 per GPU, resulting in 3.8 total iterations. All models were trained in the same pytorch environment (torch 1.7, torchvision 0.8.1), on two Nvidia RTX 3090 cards. Class-wise losses were weighted to reflect the frequency of each class in their respective subsets. The complete dataset was split into roughly 3 equal parts. Two were employed for training and validation, and the third test set remained hidden to allow for future challenges similar to [48]. Further information on data availability is described in the Data Availability section at the end of the paper.

### 3.2. Evaluating Weather Conditioning

Due to the auxiliary branch being trained in a supervised manner, it has to be exposed to the variety observed in the training set. As such all clips in the dataset were evenly distributed across equally-sized training, test, and validation sets. Because this potentially allows a naive approach to generalize easily, due to the inclusion of the full variation present in the dataset, the proposed method is compared to an equally trained naive approach without the auxiliary meta-prediction branch.

To evaluate the potential impact of each of the three meta-conditions (namely, temperature, humidity, and time of day), each model was trained naively (i.e., according to the training loop described in the respective paper [2,54]), as well as with the auxiliary conditioning branch (direct and indirect conditioning for YOLO- and DETR-variants, respectively). To allow for fair optimization, each model was trained for the same amount of epochs as their respective baseline.

Likewise, we observed the performance when comparing the temperature and object size to investigate if any of the categories potentially suffered. This was conducted in order to reach a more general improvement of the system. While these correlations might not be intuitively tied to the weather, the latent representation learned could inadvertently favor certain aspects of the object distributions.

Because conditions are quantified using different metrics, the ground truth ranges varied significantly. To normalize their representation, the values were remapped so that the observed values fell roughly within the range of $[-2, 2]$. This range was chosen to avoid the network having to also learn a mapping between arbitrary ranges while keeping in line with the normalization that is conducted internally in the networks (which is performed to avoid unstable variances in the activations [32,33,50]).

### 3.3. Accuracy

Table 1 details the overall mAP and MR for all of their models across the validation set. This is used as a similar metric of overall object detection performance to what is commonly performed for other object detection datasets, as well as to retain a fair comparison with the original LTD dataset evaluation [1]. Additionally, Table 2 details the MAE of the auxiliary prediction branch, as well as the Std. of the the prediction error. This is listed to provide insight into the performance of the auxiliary branch. We report on the accuracy following the common VOC and MS COCO dataset metrics (particularly MS COCO, which also delineates between objects sizes). Thus, in Table 1 $mAP_{VOC}$ denotes mAP where IoU is at least 0.5, and $mAP_{COCO}$ denotes the mAP at varying IoUs (i.e., {0.50, 0.55, 0.60, ..., 0.95}). $mAP_L$, $mAP_M$, and $mAP_S$ denote the mAP of the objects with {$area < 32^2$, $area > 32^2 < 96^2$, and $area > 96^2$}, respectively.

As can be seen in Table 1, the baseline models that are naively trained without any auxiliary guidance tend to perform better on primary task metrics (mAP); however, weather-conditioned variants (particularly temperature variants) display reduced miss rates, indicating that—while their accuracy is generally lower—they recognize more objects than their baseline counterpart.

**Table 1.** In this table, the mAP and MR of the direct (YOLOv5) and indirect conditioning (DETR) variants are detailed. Highlighted with **bold** is the best performing model across all of the models, and highlighted with <u>underline</u> is the best performing model for a given architecture. Temperature, humidity, and time of day are abbreviated as Temp., Hum., and ToD, respectively.

| Model | $mAP_{voc}$ | $mAP_{coco}$ | $mAP_L$ | $mAP_M$ | $mAP_S$ | MR |
|---|---|---|---|---|---|---|
| YOLOv5 (Baseline) | **0.604** | **0.465** | **0.825** | **0.640** | 0.491 | 0.342 |
| YOLOv5 (Pretrain) | 0.600 | 0.454 | 0.831 | 0.621 | 0.489 | 0.324 |
| YOLOv5 (Temp.) | 0.584 | 0.410 | 0.796 | 0.590 | 0.468 | <u>0.322</u> |
| YOLOv5 (Hum.) | 0.493 | 0.293 | 0.675 | 0.560 | 0.268 | 0.357 |
| YOLOv5 (ToD) | 0.549 | 0.439 | 0.805 | 0.566 | 0.431 | 0.356 |
| DN-DETR Baseline | <u>0.378</u> | <u>0.348</u> | <u>0.123</u> | <u>0.344</u> | 0.563 | 0.421 |
| DN-DETR (Temp.) | 0.225 | 0.148 | 0.100 | 0.190 | **0.682** | <u>0.389</u> |
| DN-DETR (Hum.) | 0.191 | 0.132 | 0.100 | 0.160 | 0.671 | 0.415 |
| DN-DETR (ToD) | 0.219 | 0.142 | 0.00 | 0.169 | 0.661 | 0.410 |
| Def. DETR Baseline | <u>0.332</u> | <u>0.202</u> | <u>0.005</u> | <u>0.051</u> | <u>0.637</u> | 0.383 |
| Def. DETR (Temp.) | 0.297 | 0.184 | 0.001 | 0.045 | 0.620 | **0.351** |
| Def. DETR (Hum.) | 0.213 | 0.114 | 0.000 | 0.020 | 0.517 | 0.416 |
| Def. DETR (ToD) | 0.289 | 0.178 | 0.001 | 0.040 | 0.619 | 0.395 |

**Table 2.** In terms of the accuracy of the predicted auxiliary prediction value, *Dir.* and *Indir.* denote the direct and indirect conditioning models, respectively. Meanwhile, the model row denotes the variant used.

| | Model | MAE | MAPE | Std. |
|---|---|---|---|---|
| Dir. | Temperature | 7.1 | 18% | 3.7 |
| | Humidity | 18.9 | 49% | 9.4 |
| | Time of Day | 7.3 | 19% | 7.1 |
| Indir. | Temperature | 5.1 | 14% | 2.9 |
| | Humidity | 15.3 | 43% | 8.9 |
| | Time of Day | 8.3 | 22% | 7.9 |

*3.4. Accuracy Compared to Weather*

To evaluate the impact of the conditioning branch on the performance with respect to the different weather conditions used for auxiliary prediction and optimization, Figures 7–9 detail the relation between mAP and the three meta-variables chosen (namely temperature, humidity, and time of day). Visual examples to accompany the accuracy overview of Tables 1 and 2 can be seen in Figures 10 and 11 (ground truth labels and the image without bounding boxes can be found Figure A3), while visualizations of accuracy with respect to the different weather variables can be seen in Figures 7–9.

In Figure 7, it can be observed that training models with a temperature-focused auxiliary branch do not change the performance of said model in any significant way (other than a generally lowered mAP). It can be seen that all models follow a curve that is similar to the distribution of samples seen in Figure 4b, and it can be expected that this is happening as the model optimization is simpler when regressing to the mean of the dataset. In addition, it can be observed that the indirect conditioning method is generally more agnostic to variation in the meta-variables. Similarly to the temperature-focused auxiliary branch, humidity- and time-of-day conditioning do not seem to improve the overall performance of the models. However, interestingly, the models seem to be generally agnostic to the distribution of samples (shown in Figure 4a,c). This indicates that the model has trouble extracting meaningful information with regard to the auxiliary optimization task. This is also present in Table 2, which is where it can be seen that the networks have difficulty with accurately predicting their respective weather conditions (specifically humidity and the time of day), whereas temperature prediction is rather accurate and falls close to the acceptable deviation of the $L1_e$ loss.

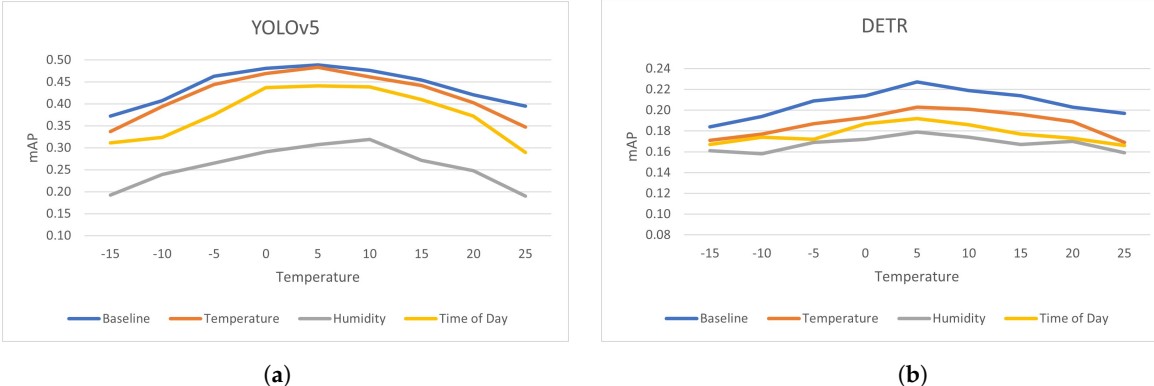

(**a**)                                                                     (**b**)

**Figure 7.** mAP of (**a**) YOLO and (**b**) DETR variants, with respect to temperature of samples in the test-set. Please note the Y-axis ranges differ between architectures.

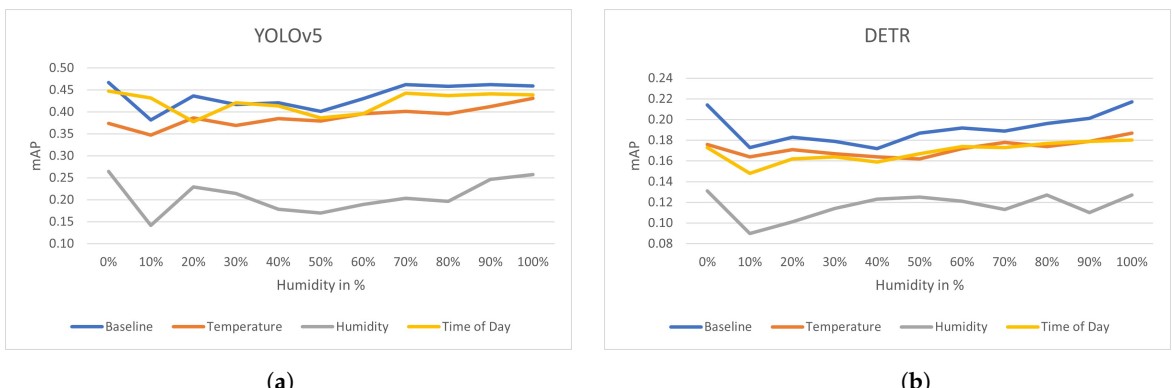

(**a**)                                                                     (**b**)

**Figure 8.** mAP of (**a**) YOLO and (**b**) DETR variants, with respect to the humidity of samples in the test-set. Please note the Y-axis ranges differ between architectures.

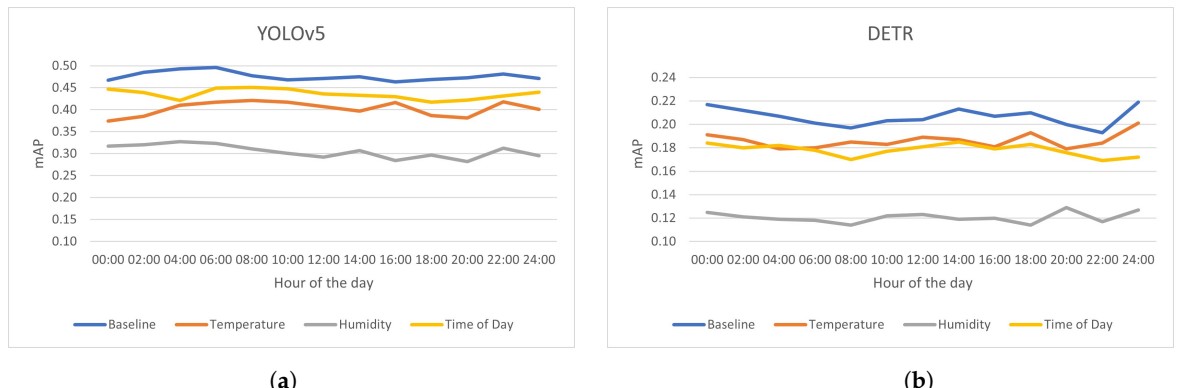

(**a**)                                                                     (**b**)

**Figure 9.** mAP of (**a**) YOLO and (**b**) DETR variants, with respect to time-of-day of the samples in the test-set. Please note the Y-axis ranges differ between architectures.

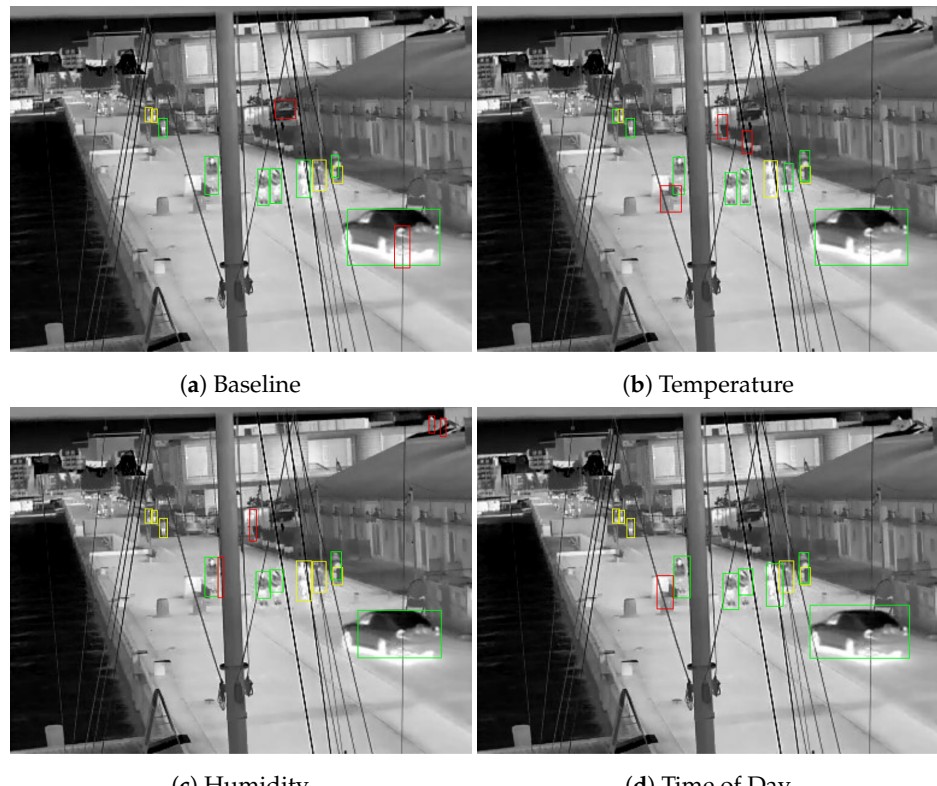

**(a)** Baseline      **(b)** Temperature

**(c)** Humidity      **(d)** Time of Day

**Figure 10.** Example of direct conditioning performance for each conditioned model. Bounding boxes marked in green, red, yellow are considered true positives, false positives, and false negatives, respectively.

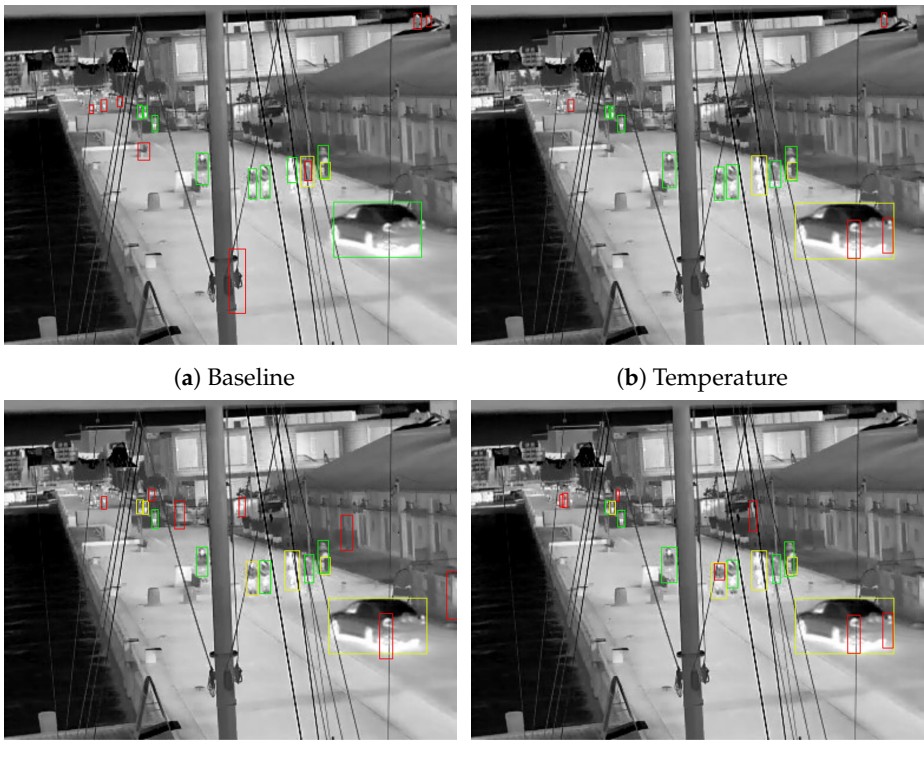

**(a)** Baseline      **(b)** Temperature

**(c)** Humidity      **(d)** Time of Day

**Figure 11.** Example of indirect conditioning performance for each conditioned model. Bounding boxes marked in green, red, yellow are considered true positives, false positives, and false negatives, respectively.

## 4. Discussion

Previous work has shown that traditional Convolutional Neural Networks (CNNs) is able to predict weather categories, and—in some contexts—it can help guide the network to be aware of the distribution a given sample belongs to and to adjust accordingly. While this has not been shown to increase the accuracy in terms of mAP, it has been shown to decrease false negative predictions. Thermal images with significant concept drift could introduce artifacts that would look appropriate for a given object in one distribution but would be undesired for another. Intuitively, during training, the model would either adjust to over-predict (i.e., increased false positives), or under-predict (i.e., increased false negatives) when concept drift occurred. Essentially, the model was tasked with learning an unknown set of distributions, and it was optimized toward learning to recognize patterns common to the mean of the cumulative distribution. Therefore, one could hypothesize that guiding the network toward being aware of a variable correlated with the observed concept drift would allow the network to potentially establish connections between the conditioning representation and the semantic representation used for object prediction.

While it can be observed in Table 1 that the mAP scores do not improve over baseline when conditioned with the auxiliary branch, the change in MR indicates that the auxiliary branch is enforcing a signal that relates to the auxiliary task. In particular, the temperature-conditioned variant managed to detect objects that the baseline failed to detect. However, the weather-conditioned method also produced an increase in false positives. Because the visual appearance changes were gradual, resulting in lower accuracy, the weather-conditioned model potentially learned a more varied representation of given objects, which allowed it to detect more objects at the cost of false activations in other places. Additionally, it can be observed (in Figure 7) that the transformer-based model performed significantly more uniformly across temperatures. However, it was not certain that this is entirely an aspect of the auxiliary predictive branch or due to the nature of the transformers' input-dependent attention. Another surprising detail can be found in Table 1, which showed that the DETR-variants, in general, seemed to work really well on small objects, which is counter to what is observed in the original and subsequent papers [32,49,54]. We might reasonably conclude that this is partially due to the decoder module, which has a fixed amount of learnable query tokens and should naturally converge toward spatial and latent features that are the most prominent, i.e., the person class. Initially, an experiment was conducted with regard to the number of queries to produce, and while increasing them drastically (300->600) would improve performance by roughly 0.3%, the performance would also increase significantly. The increased amount of query tokens could have been kept as a baseline. However, for the sake of keeping baseline models (i.e., YOLOv5 and Deformable DETR) somewhat comparable with other work, the hyperparameters described in their respective repositories and paper were kept.

Previous work, namely task-conditioned YOLO [2], evaluated performance on the KAIST dataset, which has two distinct thermal distributions (day and night clips). Our approach assumed that, during real-world deployment, a more continual representation would be present and thus could be learned. However, the results indicated that, while a signal can be extracted, there still is an underlying need for an additional proxy to guide the formation of sub-distributions of similar visual appearance. The challenge of regressing an exact numerical value for weather conditions can also be observed in the field of weather forecasting [8,12,56], where the choice of optimization strategy varies despite the task being similar, and the non-linear increase in loss the further the prediction strays from the ground truth is quite prominent. Adjacent computer vision approaches, however, tend to leverage a binned classification approach [9,11,13], which can simplify the prediction tasks significantly. While the loss proposed in Section 2.2 attempts to find a good medium between binned and continuous prediction, it perhaps provides an unstable signal due to the exponential growth when exceeding the accepted deviance $k$. While the proposed conditioning methods force the object detectors to take visual information related to weather into account, it does not seem to directly improve primary metrics such as mAP;

thus, explicitly modeling the weather in this manner could be considered ineffective when optimizing for mAP.

The appearance shift induced when the thermal camera calibrates to adjust the internal thermograph, perhaps, induces some noise into the signal, thus making it difficult to learn a robust approximation of the signal. It could be the visual noise induced partially obfuscates clear delineations between visual groups of visually similar samples, thus resulting in a regression to the mean being the simplest convergence or perhaps the optimal solution for the downstream task. In such a situation naively training without the auxiliary branch would be the optimal solution if the goal is simply to optimize accuracy. While the auxiliary task does seem to induce noise, both methods (direct and indirect conditioning) seem to also somewhat guide the network toward containing a more continual representation, as seen by the reduction in MR. Perhaps trying to construct distributions as a series of $k$ overlapping distributions, and leveraging a model-soup [57]-style approach could provide a more distinct learning of each sub-distribution, while still achieving a generalized model of all distributions.

## 5. Conclusions

Thermal concept drift poses a challenging hurdle to overcome when deploying object recognition systems. Drawing from contextual clues that impact the visual appearance of the scene could be beneficial. Using auxiliary metrics to condition a network directly or indirectly does not seem to improve the overall performance of the system with regard to mAP. However, it does result in a consistent decrease in MR. While not resulting in a direct improvement, this shows that a signal can be extracted from the conditioning meta-variable, which can guide the representations learned. The difficulty in accurately modeling the objects across thermal signatures seems to prefer, similar to a naively trained baseline, representations that favor the most frequent representations; as such, this could be simply seen as a regression to the mean. However, due to the networks consistently being able to extract a signal related to the auxiliary task, it could imply that deliberately splitting the data into a set of $K$ distributions based on a combination of meta-variables or visual appearance could provide more stable guidance.

**Author Contributions:** Conceptualization, A.S.J., K.N. and S.E.; methodology, A.S.J., K.N., S.E. and T.B.M.; software, A.S.J.; validation, A.S.J.; formal analysis, A.S.J.; investigation, A.S.J.; resources, A.S.J., K.N. and T.B.M.; data curation, A.S.J. and K.N.; writing—original draft preparation, A.S.J.; writing—review and editing, A.S.J., K.N. and S.E.; visualization, A.S.J.; supervision, K.N., S.E. and T.B.M.; project administration, K.N., A.S.J. and T.B.M.; funding acquisition, K.N. All authors have read and agreed to the published version of the manuscript.

**Funding:** This research received no external funding.

**Institutional Review Board Statement:** Not Applicable.

**Informed Consent Statement:** Not Applicable.

**Data Availability Statement:** The data used for this work is partially available on Kaggle (https://www.kaggle.com/datasets/ivannikolov/longterm-thermal-drift-dataset, accessed on 11 September 2023), and the remaining object annotations will become available soon.

**Acknowledgments:** This paper was made as a part of the Milestone Research Programme at Aalborg University (MRPA) and in collaboration with Sergio Escalera at the University of Barcelona.

**Conflicts of Interest:** The authors declare that there are no conflicts of interest regarding the publication of this paper.

## Abbreviations

The following abbreviations are used in this manuscript:

| | |
|---|---|
| AP | Average Precision 6 |
| CCTV | Closed Circuit Television 3 |
| CNN | Convolutional Neural Network 14 |
| DETR | Detection Transformer 9–11, 14 |
| IoU | Intersection over Union 6, 8, 10 |
| KAIST | KAIST Multispectral Pedestrian Detection 4, 6, 7, 14, 17 |
| LTD | Long-term Thermal Drift 3–7, 10 |
| MAE | Mean Average Error 7, 10 |
| mAP | Mean Average Precision 6, 8, 10, 11, 13–15 |
| MAPE | Mean Average Percentage Error 7 |
| MR | Miss Rate 8, 10, 11, 14, 15 |
| MS COCO | Microsoft Common Objects in Context 6, 10 |
| MWD | Multi-class Weather Dataset 2 |
| RFS | Rain Fog Snow 2 |
| SotA | State of the Art 4 |
| Std. | Standard Deviation 7, 10 |
| VOC | Pascal Visual Object Challenge 6, 10 |
| YOLO | You Only Look Once 2, 4, 8, 10, 14 |

## Appendix A. Contrast-Enhanced KAIST Examples

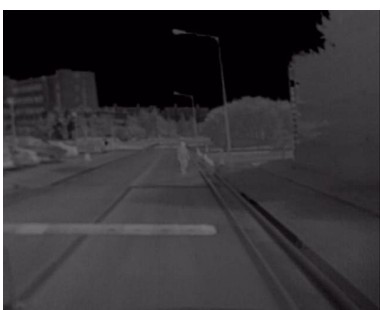

(**a**) Daytime Example.

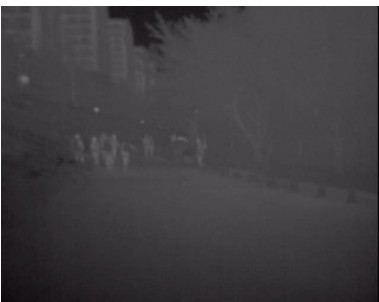

(**b**) Nighttime Example.

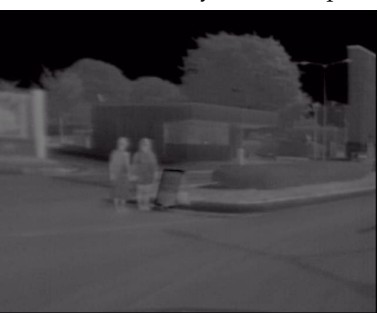

(**c**) Daytime Example.

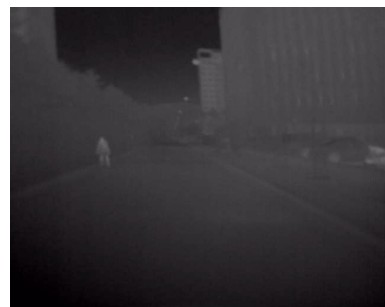

(**d**) Nighttime Example.

**Figure A1.** Contrast-enhanced variants of the examples visualized in Figure 1, showing examples of of the thermal signature of objects in the KAIST dataset.

## Appendix B. Additional Dataset Figures

*Class-Wise Object Distributions*

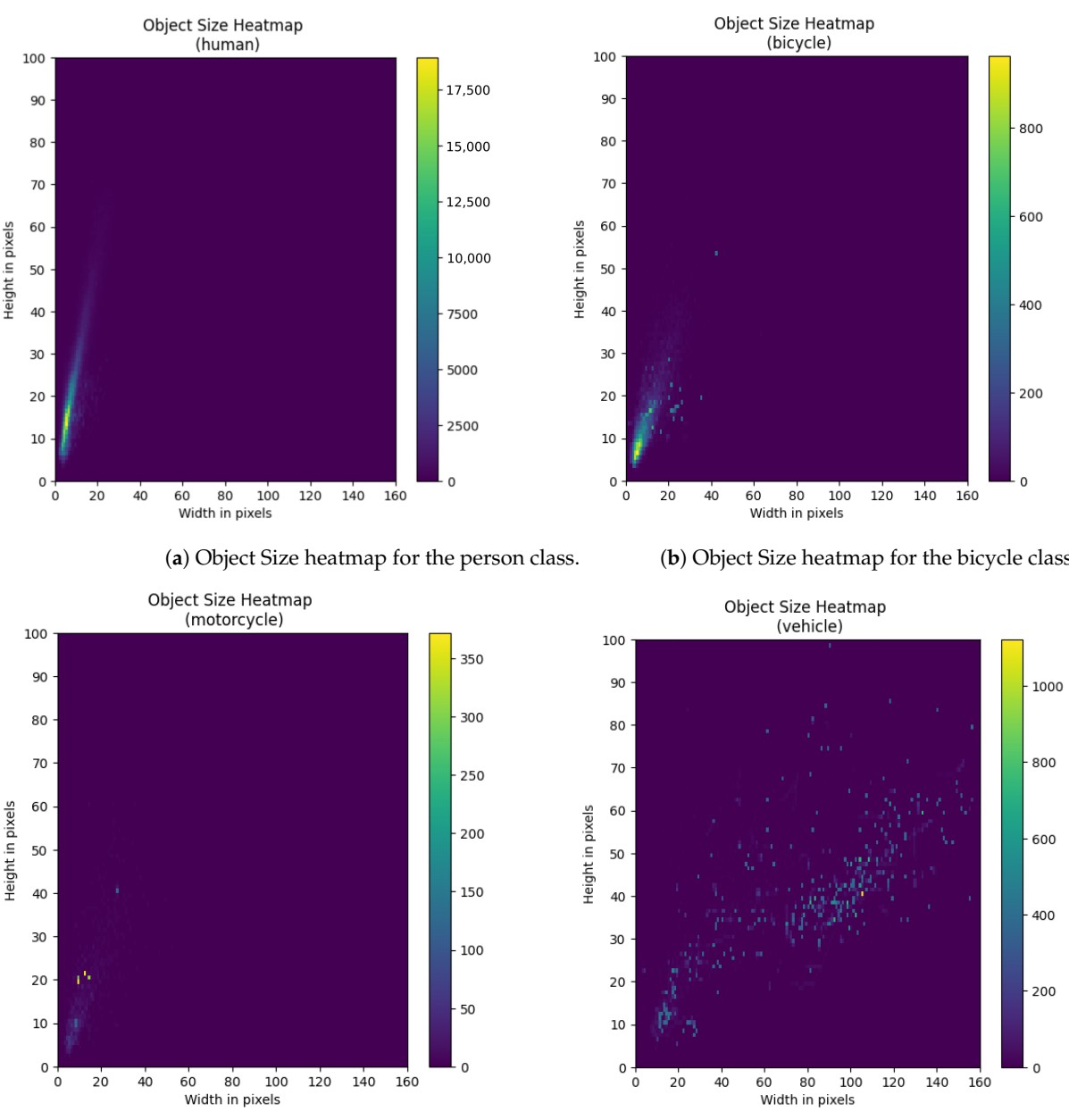

(**a**) Object Size heatmap for the person class.

(**b**) Object Size heatmap for the bicycle class.

(**c**) Object Size heatmap for the motorcycle class.

(**d**) Object Size heatmap for the vehicle class.

**Figure A2.** (**a**–**d**) show a detailed heatmap of the object size distributions for each class individually.

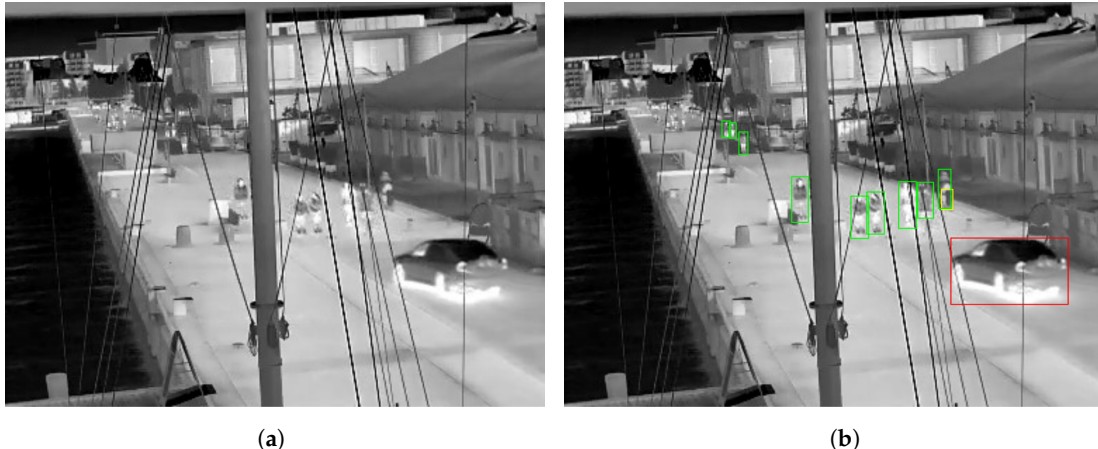

| (**a**) | (**b**) |

**Figure A3.** The example image used in Figures 10 and 11 is shown without bounding boxes (**a**) and with bounding boxes (**b**). In Figure 10 green, yellow, and red refer to person, bicycle, and vehicle classes, respectively.

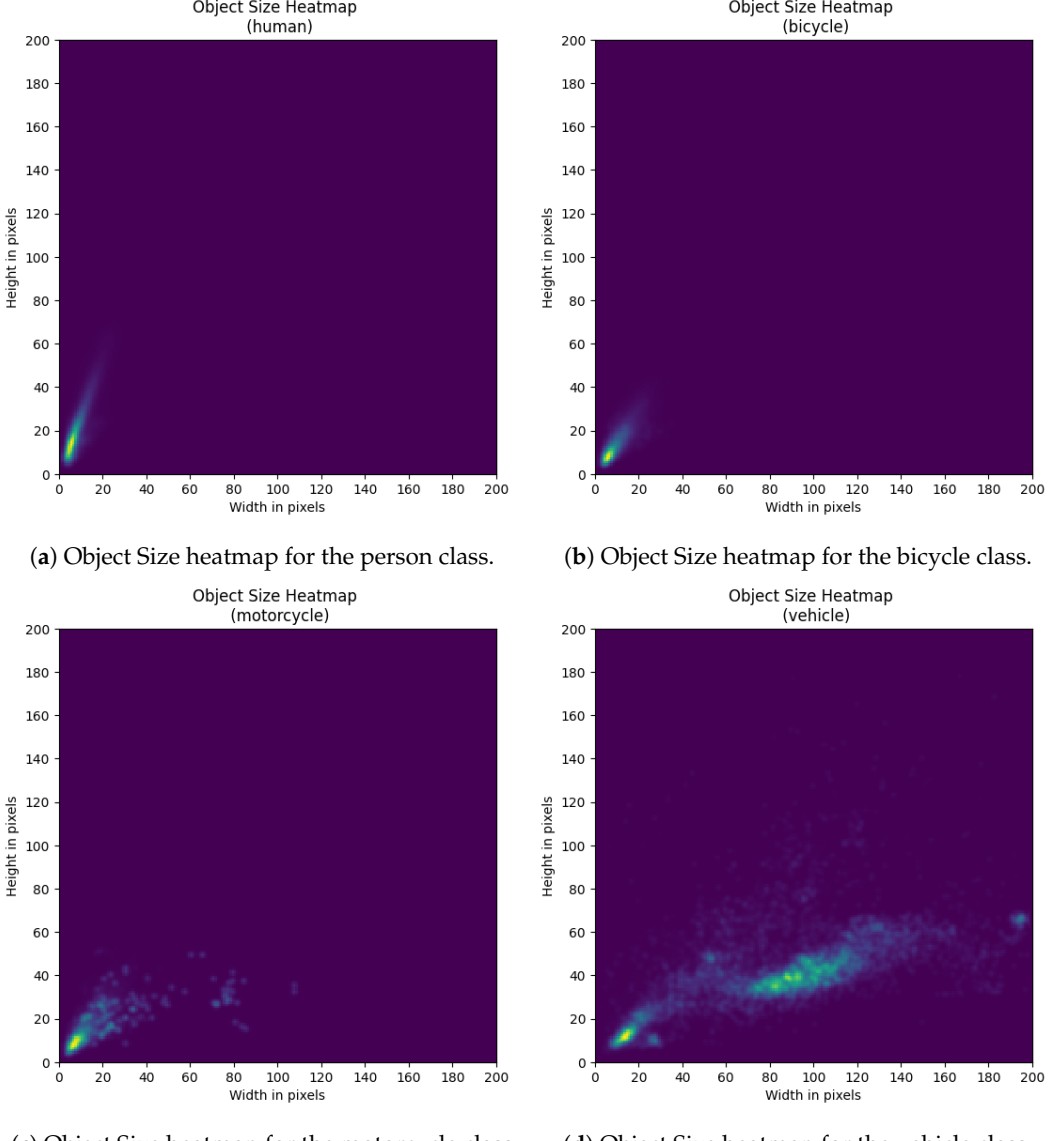

(**a**) Object Size heatmap for the person class.

(**b**) Object Size heatmap for the bicycle class.

(**c**) Object Size heatmap for the motorcycle class

(**d**) Object Size heatmap for the vehicle class.

**Figure A4.** Heatmaps of object size distributions for each class in the training set.

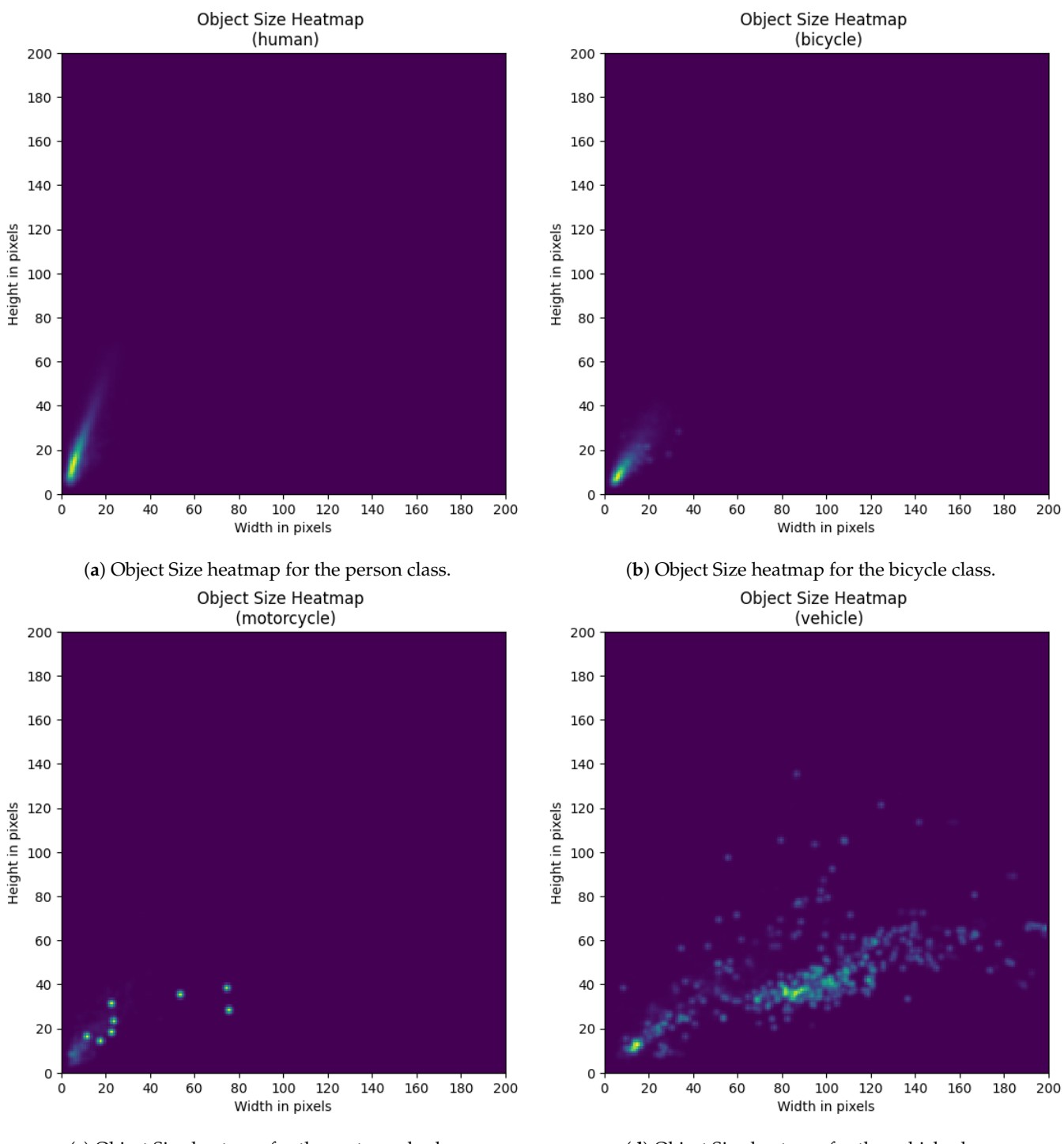

(**a**) Object Size heatmap for the person class.

(**b**) Object Size heatmap for the bicycle class.

(**c**) Object Size heatmap for the motorcycle class

(**d**) Object Size heatmap for the vehicle class.

**Figure A5.** Heatmaps of object size distributions for each class in the validation set.

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
