# Peer review of "Who Cares about the Weather? Inferring Weather Conditions for Weather-Aware Object Detection in Thermal Images"

_applsci, doi:10.3390/app131810295_

Round 1

Reviewer 1 Report

Abstract is too general. It lacks methodological details, as well as results are not properly represented (no concrete figures, achievements, etc.).

Revise all the abbreviations. They should be explained prior to their appearance in the text.

Line 149: double use of the word “namely” in the sentence.

It seems like most of the study methodology basics are grounded on the [2] Kieu, M.; Bagdanov, A.D.; Bertini, M.; Del Bimbo, A. Task-conditioned domain adaptation for pedestrian detection in thermal 421 imagery. In Proceedings of the Computer Vision–ECCV 2020: 16th European Conference, Glasgow, UK, August 23–28, 2020, 422 Proceedings, Part XXII 16. Springer, 2020, pp. 546–562. It is interesting whether there are no alternative studies on the topic, or you failed in finding additional sources?

I suggest you additionally provide mean average percentage error (MAPE) together with MAE for better understanding of the differences between the predicted and true values.

Line 340: Please, check the sentence structure.

Discussion could be extended at the expense of adding more speculation on the achievements of other scientists in this field of knowledge and pointing out the weaknesses and strong points of your own research results.

Not all the abbreviations were listed in the Acronyms section.

Reviewer 2 Report

Dear Authors,

I went through your text, which I generally appreciate, and congratulate you on the interesting topic you intend to publish. Nevertheless, while reading your manuscript, I came across numerous unclarities and errors that need solid improvements before the resubmission. 

Starting from the beginning - I'm wondering why you chose such a title. I know it should have been catchy, describing the whole idea as a compelling tagline. However, while reading your text once can expect that the question will eventually be answered. In this case, it is not. Generally, you are responsible for your text, but I really suggest reconsidering it.

The text makes an impression that it is a bit chaotic. The flag example is citing the literature in a mixed order. Please refer to the journal guidelines for details, but I can also assure you that the text will look much more transparent when you apply the rule. Moreover, citing one position several times (e.g., in lines 26, 28, 30, and then even in line 140, or 53, 54, and so on) does not look professional. The text should be concise, presenting a thought unambiguously. Generally, the entire section needs to be rebuilt in terms of that. 

Line 130: KAIST and also other acronyms need to be explained - not only as a glossary at the end of the paper but with citing appropriate literature.

Figure 1: each picture is very hard to read. I am aware that it was partially your intention to show how difficult to distinguish it was; however, I suggest making them at least more visible.

Lines 144-145: the wrong separator is used (a dot); it makes the whole passage confusing.

Line 149: the sentence is unclear (is it a repetition?).

Lines 151 and 153: "as" should begin with a small letter.

Figure 3: as the left picture is understandable, the right one is not. The description beneath is also vague so one can have difficulties with recognizing what such ellipses really mean (and what they refer to).

Line 183: the formula appears all of a sudden. It needs comments, parameter description, and citing relevant literature if it is not the authors' finding.

Line 197: again - coming back to [2] makes the whole text chaotic.

Lines 204-209: each acronym and parameter mentioned here must be explained, commented on, and equipped with the relevant citation.

Line 215: we are currently in Section 2.3! I don't understand your intention.

Figure 5: it is hard to guess why the scheme presented here is relevant. What is more, the parameters need description and comments if they are so important to be presented on the graph. The same refers to Figure 6.

Tables 1 and 2 are wrongly entitled. Please refer to the guidelines for the authors to get more details.

The description under Table 1 (should be above) contains comments and parameters that should be precisely and understandably explained in the text. A title should contain general information and should it be too complex to be presented in one line, it needs to be discussed separately.

Line 240 refers to Figure 6 whereas it is placed in another, forthcoming section. Then, the figure itself is merely tied with Section 3.1. It all makes the text confusing. 

Line 300: the information that the figures come from appendices is desirable here.

Lines 373-375: repetitions.

Summing up, I recommend you rethink the whole text and try to write it more concisely. Once the corrections are introduced, the manuscript should be resubmitted for another review cycle. Good luck.

The English language is generally understandable and - in most cases - correct. However, the text contains many repetitions, vague passages, and stylistic unclarities. What is more, there are numerous lack of spaces between the text and citations. The text makes me consider that the paper was prepared rapidly, without a thorough check and in-depth proofreading. I am sorry to say that, but the manuscript is a bit chaotic and difficult to follow. I suggest providing an independent correction and doing a proofreading once again.

Round 2

Reviewer 2 Report

Dear Authors,

Thank you for your revised manuscript. I read it carefully and want to thank you for considering many of my suggestions, especially those referring to the linguistic items. Also, many unclarities are now erased, so, in my opinion, your text looks much more transparent now. Nevertheless, I've noticed some minor mistakes that still need improvement. 

First of all, the response you attached as a cover letter is very hard to read because you mixed all your answers for both reviewers inside the letter that should be addressed to the editors, not to me. As a reviewer, I would rather expect precise answers to my concerns - not necessarily an elaborate addressed to everyone involved in the reviewing process. What's more, you answered a question about using a wrong parameter that I didn't even ask (probably another reviewer did). Also, your response to reviewer 2 contains some types, at least in the PDF you have sent. It probably comes out from a bit turbulent form of writing you prefer.

Hence, in my opinion, you missed some answers I expected, at least to know. For future publishing in Applied Sciences, I advise you to follow the strict guidelines of MDPI including also the form of providing the answers to the reviewers, text formatting, and many more. It will save our common time together with some potential misunderstandings. 

Below, please find my additional questions:

1) Starting from the beginning - I'm wondering why you chose such a title. I know it should have been catchy, describing the whole idea as a compelling tagline. However, while reading your text, one can expect that the question will eventually be answered. In this case, it is not. Could you comment on it, please?

2) Figure 1: Each picture is very hard to read. I am aware that it was partially your intention to show how difficult to distinguish it was; however, I suggest making them at least more visible.

3) Formula 1: Is the formula your own finding or just a modification of an existing one? It needs to be commented.

4) Figures 5 and 6 contain symbols and abbreviations ('sigmas', 'betas', and many more) that are nowhere explained.

5) Figures 10 and 11: One elementary comment - if your values are spanned between 0.1 and 0.2 (roughly) there is no need to select an oversized scale up to 0.5 because the majority of your chart looks empty whereas all color lines overlap each other. I recommend you shorten the range and present the accuracy distribution more precisely.

Please consider all the suggestions and submit your text for another - hopefully - the last check.  I wish you good luck!

English is more transparent now; however, some minor mistakes still need corrections. I recommend reading the manuscript once again but very carefully regarding potential typoes, commas, spaces, etc. (e.g., lines 189-190 'could serves', line 266, and so on).

Round 3

Reviewer 2 Report

Dear Authors,

Thank you very much for your reply. At the time, I was satisfied and appreciated your explanations. I also accept all the technical improvements you provided to your text. In my opinion, it looks professional now and is much more informative than before.

Regarding that, I don't have any other concerns and recommend your work for publishing.  

English language has been improved and corrected respectively, so I don't have any other objections.